# Neural Set Function Extensions:
# Learning with Discrete Functions in High Dimensions

**Nikolaos Karalias**[*]
EPFL
nikolaos.karalias@epfl.ch

**Joshua Robinson**[*]
MIT CSAIL
joshrob@mit.edu

**Andreas Loukas**
Prescient Design, Genentech, Roche
andreas.loukas@roche.com

**Stefanie Jegelka**
MIT CSAIL
stefje@csail.mit.edu

## Abstract

Integrating functions on discrete domains into neural networks is key to developing their capability to reason about discrete objects. But, discrete domains are (I) not naturally amenable to gradient-based optimization, and (II) incompatible with deep learning architectures that rely on representations in high-dimensional vector spaces. In this work, we address both difficulties for set functions, which capture many important discrete problems. First, we develop a framework for extending set functions onto low-dimensional continuous domains, where many extensions are naturally defined. Our framework subsumes many well-known extensions as special cases. Second, to avoid undesirable low-dimensional neural network bottlenecks, we convert low-dimensional extensions into representations in high-dimensional spaces, taking inspiration from the success of semidefinite programs for combinatorial optimization. Empirically, we observe benefits of our extensions for unsupervised neural combinatorial optimization, in particular with high-dimensional representations.

## 1 Introduction

While neural networks are highly effective at solving tasks grounded in basic perception (Chen et al., 2020; Vaswani et al., 2017), discrete algorithmic and combinatorial tasks such as partitioning graphs, and finding optimal routes or shortest paths have proven more challenging. This is, in part, due to the difficulty of integrating discrete operations into neural network architectures (Battaglia et al., 2018; Bengio et al., 2021; Cappart et al., 2021a). One immediate difficulty with functions on discrete spaces is that they are not amenable to standard gradient-based training. Another is that discrete functions are typically expressed in terms of scalar (e.g., Boolean) variables for each item (e.g., node, edge to be selected), in contrast to the high-dimensional and continuous nature of neural networks' internal representations. A natural approach to addressing these challenges is to carefully choose a function on a continuous domain that *extends* the discrete function, and can be used as a drop-in replacement.

There are several important desiderata that such an extension should satisfy in order to be suited to neural network training. First, an extension should be valid, i.e., agree with the discrete function on discrete points. It should also be amenable to gradient-based optimization, and should avoid introducing spurious minima. Beyond these requirements, there is one additional critical consideration. In both machine learning and optimization, it has been observed that high-dimensional representations can make problems "easier". For instance, neural networks rely on high-dimensional internal

---

[*]Equal contribution.

36th Conference on Neural Information Processing Systems (NeurIPS 2022).

representations for representational power and to allow information to flow through gradients, and performance suffers considerably when undesirable low-dimensional bottlenecks are introduced into network architectures (Belkin et al., 2019; Veličković & Blundell, 2021). In optimization, *lifting* to higher-dimensional spaces can make the problem more well-behaved (Goemans & Williamson, 1995; Shawe-Taylor et al., 2004; Du et al., 2018). Therefore, extending discrete functions to *high-dimensional* domains may be critical to the effectiveness of the resulting learning process, yet remains largely an open problem.

With those considerations in mind, we propose a framework for constructing extensions of discrete set functions onto high-dimensional continuous spaces. The core idea is to view a continuous point $\mathbf{x}$ in space as an expectation over a distribution (that depends on $\mathbf{x}$) supported on a few carefully chosen discrete points, to retain tractability. To evaluate the discrete function at $\mathbf{x}$, we compute the expected value of the set function over this distribution. The method resulting from a principled formalization of this idea is computationally efficient and addresses the key challenges of building continuous extensions. Namely, our extensions allow gradient-based optimization and address the dimensionality concerns, allowing any function on sets to be used as a computation step in a neural network.

First, to enable gradient computations, we present a method based on a linear programming (LP) relaxation for constructing extensions on continuous domains where exact gradients can be computed using standard automatic differentiation software (Abadi et al., 2016; Bastien et al., 2012; Paszke et al., 2019). Our approach allows task-specific considerations (e.g., a cardinalilty constraint) to be built into the extension design. While our initial LP formulation handles gradients, and is a natural formulation for explicitly building extensions, it replaces discrete Booleans with scalars in the unit interval $[0, 1]$, and hence does not yet address potential dimensionality bottlenecks. Second, to enable higher-dimensional representations, we take inspiration from classical SDP relaxations, such as the celebrated Goemans-Williamson maximum cut algorithm (Goemans & Williamson, 1995), which recast low-dimensional problems in high-dimensions. Specifically, our key contribution is to develop an SDP analog of our original LP formulation, and show how to *lift* LP-based extensions into a corresponding high-dimensional SDP-based extensions. Our general procedure for lifting low-dimensional representations into higher dimensions aligns with the neural algorithmic reasoning blueprint (Veličković & Blundell, 2021), and suggests that classical techniques such as SDPs may be effective tools for combining deep learning with algorithmic processes more generally.

## 2 Problem Setup

Consider a ground set $[n] = \{1, \ldots, n\}$ and an arbitrary function $f : 2^{[n]} \to \mathbb{R} \cup \{\infty\}$ defined on subsets of $[n]$. For instance, $f$ could determine if a set of nodes or edges in a graph has some structural property, such as being a path, tree, clique, or independent set (Bello et al., 2016; Cappart et al., 2021a). Our aim is to build neural networks that use such discrete functions $f$ as an intermediate layer or loss. In order to produce a model that is trainable using standard auto-differentiation software, we consider a continuous domain $\mathcal{X}$ onto which we would like to extend $f$, with sets embedded into $\mathcal{X}$ via an injective map $e : 2^{[n]} \to \mathcal{X}$. For instance, when $\mathcal{X} = [0, 1]^n$ we may take $e(S) = \mathbf{1}_S$, the Boolean vector whose $i$th entry is 1 if $i \in S$, and 0 otherwise. Our approach is to design an extension

$$\mathfrak{F} : \mathcal{X} \to \mathbb{R}$$

of $f$ and consider the neural network $\mathrm{NN}_2 \circ \mathfrak{F} \circ \mathrm{NN}_1$ (if $f$ is used as a loss, $\mathrm{NN}_2$ is simply the identity). To ensure that the extension is *valid* and amenable to automatic differentiation, we require that 1) it agrees with $f$ on all discrete points: $\mathfrak{F}(e(S)) = f(S)$ for all $S \subseteq [n]$ with $f(S) < \infty$, and 2) $\mathfrak{F}$ is continuous.

There is a rich existing literature on extensions of functions on discrete domains, particularly in the context of discrete optimization (Lovász, 1983; Grötschel et al., 1981; Calinescu et al., 2011; Vondrák, 2008; Bach, 2019; Obozinski & Bach, 2012; Tawarmalani & Sahinidis, 2002). These works provide promising tools to reach our goal of neural network training. Building on these, our method is the first to use semi-definite programming (SDP) to combine neural networks with set functions. There are, however, different considerations in the neural network setting as compared to optimization. The optimization literature often focuses on a class of set functions and aims to build extensions with desirable optimization properties, particularly convexity. We do not focus on convexity, aiming instead to develop a formalism that is as flexible as possible. Doing so maximizes the applicability of our method, and allows extensions adapted to task-specific desiderata (see Section 3.1).

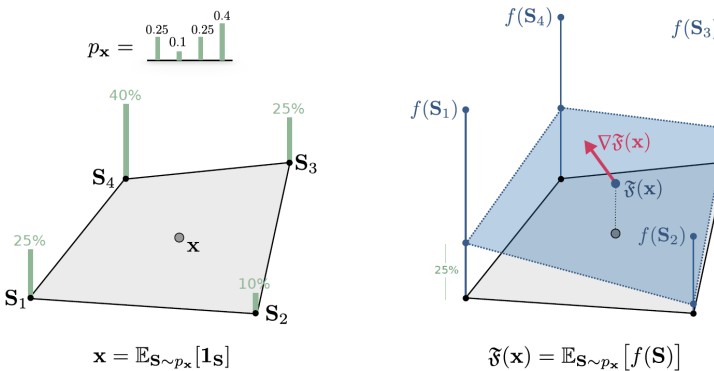

$$\mathbf{x} = \mathbb{E}_{\mathbf{S} \sim p_{\mathbf{x}}}[\mathbf{1}_{\mathbf{S}}] \qquad\qquad \mathfrak{F}(\mathbf{x}) = \mathbb{E}_{\mathbf{S} \sim p_{\mathbf{x}}}[f(\mathbf{S})]$$

Figure 1: **SFEs:** Fractional points $\mathbf{x}$ are reinterpreted as expectations $\mathbf{x} = \mathbb{E}_{S \sim p_{\mathbf{x}}}[\mathbf{1}_S]$ over the distribution $p_{\mathbf{x}}(S)$ on sets. A value is assigned at $\mathbf{x}$ by exchanging the order of $f$ and the expectation: $\mathfrak{F}(\mathbf{x})_{S \sim p_{\mathbf{x}}}[f(S)]$. Unlike $f$, the extension $\mathfrak{F}$ is amenable to gradient-based optimization.

## 3   Scalar Set Function Extensions

We start by presenting a general framework for extending set functions onto $\mathcal{X} = [0,1]^n$, where a set $S \subseteq [n]$ is viewed as the Boolean indicator vector $e(S) = \mathbf{1}_S \in \{0,1\}^n$ whose $i$th entry is 1 if $i \in S$ and 0 otherwise. We call extensions onto $[0,1]^n$ *scalar* since each item $i$ is represented by a single scalar value—the $i$th coordinate of $\mathbf{x} \in \mathcal{X}$. These scalar extensions will become the core building blocks in developing high-dimensional extensions in Section 4.

A classical approach to extending discrete functions on sets represented as Boolean indicator vectors $\mathbf{1}_S$ is by computing the convex-envelope, i.e., the point-wise supremum over linear functions that lower bound $f$ (Falk & Hoffman, 1976; Bach, 2019). Doing so yields a convex function whose value at a point $\mathbf{x} \in [0,1]^n$ is the solution of the following linear program (LP):

$$\widetilde{\mathfrak{F}}(\mathbf{x}) = \max_{\mathbf{z},b \in \mathbb{R}^n \times \mathbb{R}} \{\mathbf{x}^\top \mathbf{z} + b\} \text{ subject to } \mathbf{1}_S^\top \mathbf{z} + b \le f(S) \text{ for all } S \subseteq [n]. \qquad \text{(primal LP)}$$

The set $\mathcal{P}_f$ of all feasible solutions $(\mathbf{z}, b)$ is known as the *(canonical) polyhedron of $f$* (Obozinski & Bach, 2012) and can be seen to be non-empty by taking the coordinates of $\mathbf{z}$ to be sufficiently small (possibly negative). Variants of this optimization program are frequently encountered in the theory of matroids and submodular functions (Edmonds, 2003) where $\mathcal{P}_f$ is commonly known as the *submodular polyhedron* (see Appendix A for an extended discussion). By strong duality, we may solve the primal LP by instead solving its dual:

$$\widetilde{\mathfrak{F}}(\mathbf{x}) = \min_{\{y_S \ge 0\}_{S \subseteq [n]}} \sum_{S \subseteq [n]} y_S f(S) \text{ subject to } \sum_{S \subseteq [n]} y_S \mathbf{1}_S = \mathbf{x}, \ \sum_{S \subseteq [n]} y_S = 1, \ \text{for all } S \subseteq [n],$$
$$\text{(dual LP)}$$

whose optimal value is the same as the primal LP. The dual LP is always feasible (see e.g., the Lovász extension in Section 3.1). However, $\widetilde{\mathfrak{F}}$ does not necessarily agree with $f$ on discrete points in general, unless the function is convex-extensible (Murota, 1998).

To address this important missing piece, we relax our goal from solving the dual LP to instead seeking a *feasible* solution to the dual LP that *is* an extension of $f$. Since the dual LP is defined for a fixed $\mathbf{x}$, a feasible solution must be a function $y_S = p_{\mathbf{x}}(S)$ of $\mathbf{x}$. If $p_{\mathbf{x}}$ were to be continuous and a.e. differentiable in $\mathbf{x}$ then the value $\sum_S p_{\mathbf{x}}(S) f(S)$ attained by the dual LP would also be continuous and a.e. differentiable in $\mathbf{x}$ since gradients flow through the coefficients $y_S = p_{\mathbf{x}}(S)$, while $f(S)$ is treated as a constant in $\mathbf{x}$. This leads us to the following definition:

**Definition** (Scalar SFE)**.** A scalar SFE $\mathfrak{F}$ of $f$ is defined at a point $\mathbf{x} \in [0,1]^n$ by coefficients $p_{\mathbf{x}}(S)$ such that $y_S = p_{\mathbf{x}}(S)$ is a feasible solution to the dual LP. The extension value is given by

$$\mathfrak{F}(\mathbf{x}) \triangleq \sum_{S \subseteq [n]} p_{\mathbf{x}}(S) f(S)$$

and we require the following properties to hold for all $S \subseteq [n]$: 1) $p_{\mathbf{x}}(S)$ is a continuous function of $\mathbf{x}$ and 2) $\mathfrak{F}(\mathbf{1}_S) = f(S)$ for all $S \subseteq [n]$.

Efficient evaluation of $\mathfrak{F}$ requires that $p_{\mathbf{x}}(S)$ is supported on a small collection of carefully chosen sets $S$. This choice is a key inductive bias of the extension, and Section 3.1 gives many examples with only $O(n)$ non-zero coefficients. Examples include well-known extensions, such as the Lovász extension, as well as a number of novel extensions, illustrating the versatility of the SFE framework.

Thanks to the constraint $\sum_S y_S = 1$ in the dual LP, scalar SFEs have a natural probabilistic interpretation. An SFE is defined by a probability distribution $p_{\mathbf{x}}$ such that fractional points $\mathbf{x}$ can be written as an expectation $\mathbb{E}_{S \sim p_{\mathbf{x}}}[\mathbf{1}_S] = \mathbf{x}$ over discrete points using $p_{\mathbf{x}}$. The extension itself can be viewed as arising from exchanging $f$ and the expectation operation: $\mathfrak{F}(\mathbf{x}) = \mathbb{E}_{S \sim p_{\mathbf{x}}}[f(S)]$. This interpretation is summarized in Figure 1.

Scalar SFEs also enjoy the property of not introducing any spurious minima. That is, the minima of $\mathfrak{F}$ coincide with the minima of $f$ up to convex combinations. This property is especially important when training models of the form $f \circ \mathrm{NN}_1$ (i.e., $f$ is a loss function) since $\mathfrak{F}$ will guide the network $\mathrm{NN}_1$ towards the same solutions as $f$.

**Proposition 1** (Scalar SFEs have no bad minima). If $\mathfrak{F}$ is a scalar SFE of $f$ then:

1. $\min_{\mathbf{x} \in \mathcal{X}} \mathfrak{F}(\mathbf{x}) = \min_{S \subseteq [n]} f(S)$

2. $\arg\min_{\mathbf{x} \in \mathcal{X}} \mathfrak{F}(\mathbf{x}) \subseteq \mathrm{Hull}\big(\arg\min_{\mathbf{1}_S : S \subseteq [n]} f(S)\big)$

See Appendix B for proofs.

**Obtaining set solutions.** Given an architecture $\mathfrak{F} \circ \mathrm{NN}_1$ and input problem instance $G$, we often wish to produce sets as outputs at inference time. To do this, we simply compute $\mathbf{x} = \mathrm{NN}_1(G)$, and select the set $S$ in $\mathrm{supp}_S\{p_{\mathbf{x}}(S)\}$ with the smallest value $f(S)$. This can be done efficiently if, as is typically the case, the cardinality of $\mathrm{supp}_S\{p_{\mathbf{x}}(S)\}$ is small.

## 3.1 Constructing Scalar Set Function Extensions

A key characteristic of scalar SFEs is that there are many potential extensions of any given $f$. In this section, we provide examples of scalar SFEs, illustrating the capacity of the SFE framework for building knowledge about $f$ into the extension. See Appendix C for all proofs and further discussion.

**Lovász extension.** Re-indexing the coordinates of $\mathbf{x}$ so that $x_1 \geq x_2 \ldots \geq x_n$, we define $p_{\mathbf{x}}$ to be supported on the sets $S_1 \subseteq S_2 \subseteq \cdots \subseteq S_n$ with $S_i = \{1, 2, \ldots, i\}$ for $i = 1, 2, \ldots, n$. The coefficient are defined as $y_{S_i} = p_{\mathbf{x}}(S_i) := x_i - x_{i+1}$ and $p_{\mathbf{x}}(S) = 0$ for all other sets. The resulting *Lovász extension*—known as the *Choquet integral* in decision theory (Choquet, 1954; Marichal, 2000)—is a key tool in combinatorial optimization due to a seminal result: the Lovász extension is convex if and only if $f$ is submodular (Lovász, 1983), implying that submodular minimization can be solved in polynomial-time (Grötschel et al., 1981).

**Bounded cardinality Lovász extension.** A collection $\{S_i\}_{i=1}^n$ of subsets of $[n]$ can be encoded in an $n \times n$ matrix $\mathbf{S} \in \{0, 1\}^{n \times n}$ whose $i$th column is $\mathbf{1}_{S_i}$. In this notation, the dual LP constraint $\sum_{S \subseteq [n]} y_S \mathbf{1}_S = \mathbf{x}$ can be written as $\mathbf{Sp} = \mathbf{x}$, where the $i$th coordinate of $\mathbf{p}$ defines $p_{\mathbf{x}}(S_i)$. The *bounded cardinality* extension generalizes the Lovász extension to focus only on sets of cardinality at most $k \leq n$. Again, re-index $\mathbf{x}$ so that $x_1 \geq x_2 \ldots \geq x_n$. Use the first $k$ sets $S_1 \subseteq S_2 \subseteq \cdots \subseteq S_k$, where $S_i = \{1, 2, \ldots, i\}$, to populate the first $k$ columns of matrix $\mathbf{S}$. We add further $n - k$ sets: $S_{k+i} = \{j + i \mid j \in S_k\}$ for $i = 1, \ldots, n - k$, to fill the rest of $\mathbf{S}$. Finally, $p_{\mathbf{x}}(S_i)$ can be analytically calculated from $\mathbf{p} = \mathbf{S}^{-1}\mathbf{x}$, where $\mathbf{S}$ is invertible since it is a Toeplitz banded upper triangular matrix.

**Permutations and involutory extensions.** We use the same $\mathbf{S}, \mathbf{p}$ notation. Let $\mathbf{S}$ be an elementary permutation matrix. Then it is involutory, i.e., $\mathbf{SS} = \mathbf{I}$, and we may easily determine $\mathbf{p} = \mathbf{Sx}$ given $\mathbf{S}$ and $\mathbf{x}$. Note that $p_{\mathbf{x}}(S_i) = \mathbf{p}_i$ must be non-negative since $\mathbf{x}$ and $\mathbf{S}$ are non-negative entry-wise. Finally, restricting $\mathbf{x}$ to the $n$-dimensional Simplex guarantees that $\|\mathbf{p}\|_1 \leq 1$, which ensures $p_{\mathbf{x}}$ is a probability distribution (any remaining mass is placed on the empty set). The extension property can be guaranteed on singleton sets as long as the chosen permutation admits a fixed point at the argmax of $\mathbf{x}$. Any elementary permutation matrix $\mathbf{S}$ with such a fixed point yields a valid SFE.

**Singleton extension.** Consider a set function $f$ for which $f(S) = \infty$ unless $S$ has cardinality one. To ensure $\mathfrak{F}$ is finite valued, $p_{\mathbf{x}}$ must be supported only on the sets $S_i = \{i\}$, $i = 1, \ldots, n$. Assuming $\mathbf{x}$ is sorted so that $x_1 \geq x_2 \ldots \geq x_n$, define $p_{\mathbf{x}}(S_i) = x_i - x_{i+1}$. It is shown in Appendix C that this defines a scalar SFE, except for the dual LP feasibility. However, when using $\mathfrak{F}$ as a loss function, minimization drives $\mathbf{x}$ towards the minima $\min_{\mathbf{x}} \mathfrak{F}(\mathbf{x})$ which *are* dual feasible. So dual infeasibility is benign in this instance and we approach the feasible set from the outside.

**Multilinear extension.** The multilinear extension, widely used in combinatorial optimization (Calinescu et al., 2011), is supported on all sets with coefficients $p_{\mathbf{x}}(S) = \prod_{i \in S} x_i \prod_{i \notin S}(1 - x_i)$, the product distribution. In general, evaluating the multilinear extension exactly requires $2^n$ calls to $f$, but for several interesting set functions, e.g., graph cut, set cover, and facility location, it can be computed efficiently in $\widetilde{\mathcal{O}}(n^2)$ time (Iyer et al., 2014).

## 4 Neural Set Function Extensions

This section builds on the scalar SFE framework—where each item $i$ in the ground set $[n]$ is represented by a single scalar—to develop extensions that use high-dimensional embeddings to avoid introducing low-dimensional bottlenecks into neural network architectures. The core motivation that lifting problems into higher dimensions can make them easier is not unique to deep learning. For instance, it also underlies kernel methods (Shawe-Taylor et al., 2004) and the *lift-and-project* method for integer programming (Lovász & Schrijver, 1991).

Our method takes inspiration from prior successes of semi-definite programming for combinatorial optimization (Goemans & Williamson, 1995) by extending onto $\mathcal{X} = \mathbb{S}_+^n$, the set of $n \times n$ positive semi-definite (PSD) matrices. With this domain, each item is represented by a vector, not a scalar.

### 4.1 Lifting Set Function Extensions to Higher Dimensions

We embed sets into $\mathbb{S}_+^n$ via the map $e(S) = \mathbf{1}_S \mathbf{1}_S^\top$. To define extensions on this matrix domain, we translate the linear programming approach of Section 3 into an analogous SDP formulation:

$$\max_{\mathbf{Z} \succeq 0, b \in \mathbb{R}} \{\mathrm{Tr}(\mathbf{X}^\top \mathbf{Z}) + b\} \text{ subject to } \frac{1}{2}\mathrm{Tr}((\mathbf{1}_S \mathbf{1}_T^\top + \mathbf{1}_T \mathbf{1}_S^\top)\mathbf{Z}) + b \leq f(S \cap T) \text{ for } S, T \subseteq [n],$$
(primal SDP)

where we switch from lower case letters to upper case since we are now using matrices. Next, we show that this choice of primal SDP is a natural analog of the original LP that provides the right correspondences between vectors and matrices by proving that primal LP feasible solutions correspond to primal SDP feasible solutions with the same objective value (see Appendix A for a discussion on the SDP and its dual). To state the result, note that the embedding $e(S) = \mathbf{1}_S \mathbf{1}_S^\top$ is a particular case of the correspondence $\mathbf{x} \in [0, 1]^n \mapsto \sqrt{\mathbf{x}}\sqrt{\mathbf{x}}^\top$.

**Proposition 2.** (Containment of LP in SDP) For any $\mathbf{x} \in [0, 1]^n$, define $\mathbf{X} = \sqrt{\mathbf{x}}\sqrt{\mathbf{x}}^\top$ with the square-root taken entry-wise. Then, for any $(\mathbf{z}, b) \in \mathbb{R}_+^n \times \mathbb{R}$ that is primal LP feasible, the pair $(\mathbf{Z}, b)$ where $\mathbf{Z} = \mathrm{diag}(\mathbf{z})$, is primal SDP feasible and the objective values agree: $\mathrm{Tr}(\mathbf{X}^\top \mathbf{Z}) = \mathbf{z}^\top \mathbf{x}$.

Proposition 2 establishes that the primal SDP feasible set is a *spectrahedral lift* of the positive primal LP feasible set, i.e., feasible solutions of the primal LP lead to feasible solutions of the primal SDP. As with scalar SFEs, to define neural SFEs we consider the dual SDP:

$$\min_{\{y_{S,T} \geq 0\}} \sum_{S,T \subseteq [n]} y_{S,T} f(S \cap T) \text{ subject to } \mathbf{X} \preceq \sum_{S,T \subseteq [n]} \frac{1}{2} y_{S,T}(\mathbf{1}_S \mathbf{1}_T^\top + \mathbf{1}_T \mathbf{1}_S^\top) \text{ and } \sum_{S,T \subseteq [n]} y_{S,T} = 1$$
(dual SDP)

We demonstrate that for suitable $\mathbf{X}$ this SDP has feasible solutions via an explicit construction in Section 4.2. This leads us to define a neural SFE which, as with scalar SFEs, is given by a feasible solution to the dual SDP that satisfies the extension property whose coefficients are continuous in $\mathbf{X}$:

**Definition** (Neural SFE). A neural set function extension of $f$ at a point $\mathbf{X} \in \mathbb{S}_+^n$ is defined as

$$\mathfrak{F}(\mathbf{X}) \triangleq \sum_{S,T \subseteq [n]} p_{\mathbf{X}}(S, T) f(S \cap T),$$

where $y_{S,T} = p_{\mathbf{X}}(S,T)$ is a feasible solution to the dual SDP and for all $S,T \subseteq [n]$: 1) $p_{\mathbf{X}}(S,T)$ is continuous at $\mathbf{X}$ and 2) it is valid, i.e., $\mathfrak{F}(\mathbf{1}_S \mathbf{1}_S^\top) = f(S)$ for all $S \subseteq [n]$.

## 4.2 Constructing Neural Set Function Extensions

We constructed a number of explicit examples of scalar SFEs in Section 3.1. For neural SFEs we employ a different strategy. Instead of providing individual examples of neural SFEs, we develop a single recipe for converting *any* scalar SFE into a corresponding neural SFE. Doing so allows us to build on the variety of scalar SFEs and provides an additional connection between scalar and neural SFEs. In Section 5 we show the empirical superiority of neural SFEs over their scalar counterparts.

Our construction is given in the following proposition:

**Proposition 3.** Let $p_{\mathbf{x}}$ induce a scalar SFE of $f$. For $\mathbf{X} \in \mathbb{S}_+^n$, consider a decomposition $\mathbf{X} = \sum_{i=1}^n \lambda_i \mathbf{x}_i \mathbf{x}_i^\top$ and fix

$$p_{\mathbf{X}}(S,T) = \sum_{i=1}^n \lambda_i \, p_{\mathbf{x}_i}(S) p_{\mathbf{x}_i}(T) \text{ for all } S,T \subseteq [n].$$

Then, $p_{\mathbf{X}}$ defines a neural SFE $\mathfrak{F}$ at $\mathbf{X}$.

See Appendix D for proof. The choice of decomposition will give rise to different extensions. Here, we instantiate our neural extensions using the eigendecomposition of $\mathbf{X}$. Since eigenvectors may not belong to $[0,1]^n$ we reparameterize by first applying a sigmoid function before computing the scalar extension distribution $p_{\mathbf{x}}$. In practice we found that neural SFEs work just as well even without this sigmoid function—i.e., allowing scalar SFEs to be evaluated outside of $[0,1]^n$. The continuity of the neural SFE $\mathfrak{F}$ when using the eigendecomposition follows from a variant of the Davis–Kahan theorem (Yu et al., 2015), which requires the additional assumption that the eigenvalues of $\mathbf{x}$ are distinct. For efficiency, in practice we do not use all $n$ eigenvectors, and use only the $k$ with largest eigenvalue. This is justified by Figure 3, which shows that in practical applications $\mathbf{X}$ often has a rapidly decaying spectrum.

Evaluating a neural SFE requires an accessible closed-form expression, the precise form of which depends on the underlying scalar SFE. Further, from the definition of Neural SFEs we see that if a scalar SFE is supported on sets with a property that is closed under intersection (e.g., bounded cardinality), then the supporting sets of the corresponding neural SFE will also inherit that property. This implies that the neural counterparts of the Lovász, bounded cardinality Lovász, and singleton/permutation extensions have the same support as their scalar counterparts. An immediate corollary is that we can easily compute the neural counterpart of the Lovász extension which has a simple closed form:

**Corollary 1.** For $\mathbf{X} \in \mathbb{S}_+^n$ consider the eigendecomposition $\mathbf{X} = \sum_{i=1}^n \lambda_i \mathbf{x}_i \mathbf{x}_i^\top$. Let $p_{\mathbf{x}_i}$ be as in the Lovász extension: $p_{\mathbf{x}_i}(S_{ij}) = \sigma(x_{i,j}) - \sigma(x_{i,j+1})$, where $\sigma$ is the sigmoid function, and $\mathbf{x}_i$ is sorted so $x_{i,1} \geq \ldots \geq x_{i,n}$ and $S_{ij} = \{1, \ldots, j\}$, with $p_{\mathbf{x}_i}(S) = 0$ for all other sets. Then, the neural Lovász extension is:

$$\mathfrak{F}(\mathbf{X}) = \sum_{i,j=1}^n \lambda_i p_{\mathbf{x}_i}(S_{ij}) \cdot \left( p_{\mathbf{x}_i}(S_{ij}) + 2 \sum_{\ell:\ell>j} p_{\mathbf{x}_i}(S_{i\ell}) \right) \cdot f(S_{ij}).$$

**Complexity and obtaining sets as solutions.** In general, the neural SFE relies on all pairwise intersections $S \cap T$ of the scalar SFE sets, requiring $O(m^2)$ evaluations of $f$ when the scalar SFE is supported on $m$ sets. However, when the scalar SFE is supported on a family of sets that is closed under intersection—e.g., the Lovász and singleton extensions—the corresponding neural SFE requires only $O(m)$ function evaluations. Discrete solutions can be obtained efficiently by returning the best set out of all scalar SFEs $p_{\mathbf{x}_i}$.

## 5 Experiments

We experiment with SFEs as loss functions in neural network pipelines on discrete objectives arising in combinatorial and vision tasks. For combinatorial optimization, SFEs network training with a continuous version of the objective without supervision. For supervised image classification, they allow us to directly relax the training error instead of optimizing a proxy like cross entropy.

| | Maximum Clique | | | | |
| --- | --- | --- | --- | --- | --- |
| | ENZYMES | PROTEINS | IMDB-Binary | MUTAG | COLLAB |
| Straight-through (Bengio et al., 2013) | $0.725_{\pm 0.268}$ | $0.722_{\pm 0.26}$ | $0.917_{\pm 0.253}$ | $0.965_{\pm 0.162}$ | $0.856_{\pm 0.221}$ |
| Erdős (Karalias & Loukas, 2020) | $0.883_{\pm 0.156}$ | $0.905_{\pm 0.133}$ | $0.936_{\pm 0.175}$ | $1.000_{\pm 0.000}$ | $0.852_{\pm 0.212}$ |
| REINFORCE (Williams, 1992) | $0.751_{\pm 0.301}$ | $0.725_{\pm 0.285}$ | $0.881_{\pm 0.240}$ | $1.000_{\pm 0.000}$ | $0.781_{\pm 0.316}$ |
| Lovász scalar SFE | $0.723_{\pm 0.272}$ | $0.778_{\pm 0.270}$ | $0.975_{\pm 0.125}$ | $0.977_{\pm 0.125}$ | $0.855_{\pm 0.225}$ |
| Lovász neural SFE | $0.933_{\pm 0.148}$ | $0.926_{\pm 0.165}$ | $0.961_{\pm 0.143}$ | $1.000_{\pm 0.000}$ | $0.864_{\pm 0.205}$ |
| | Maximum Independent Set | | | | |
| | ENZYMES | PROTEINS | IMDB-Binary | MUTAG | COLLAB |
| Straight-through (Bengio et al., 2013) | $0.505_{\pm 0.244}$ | $0.430_{\pm 0.252}$ | $0.701_{\pm 0.252}$ | $0.721_{\pm 0.257}$ | $0.331_{\pm 0.260}$ |
| Erdős (Karalias & Loukas, 2020) | $0.821_{\pm 0.124}$ | $0.903_{\pm 0.114}$ | $0.515_{\pm 0.310}$ | $0.939_{\pm 0.069}$ | $0.886_{\pm 0.198}$ |
| REINFORCE (Williams, 1992) | $0.617_{\pm 0.214}$ | $0.579_{\pm 0.340}$ | $0.899_{\pm 0.275}$ | $0.744_{\pm 0.121}$ | $0.053_{\pm 0.164}$ |
| Lovász scalar SFE | $0.311_{\pm 0.289}$ | $0.462_{\pm 0.260}$ | $0.716_{\pm 0.269}$ | $0.737_{\pm 0.154}$ | $0.302_{\pm 0.238}$ |
| Lovász neural SFE | $0.775_{\pm 0.155}$ | $0.729_{\pm 0.205}$ | $0.679_{\pm 0.287}$ | $0.854_{\pm 0.132}$ | $0.392_{\pm 0.253}$ |

Table 1: **Unsupervised neural combinatorial optimization**: Approximation ratios for combinatorial problems. Values closer to 1 are better ($\uparrow$). Neural SFEs are competitive with other methods, and consistently improve over vector SFEs.

## 5.1 Unsupervised Neural Combinatorial Optimization

We begin by evaluating the suitability of neural SFEs for unsupervised learning of neural solvers for combinatorial optimization problems on graphs. We use the ENZYMES, PROTEINS, IMDB, MUTAG, and COLLAB datasets from the TUDatasets benchmark (Morris et al., 2020), using a 60/30/10 split for train/test/val. We test on two problems: finding maximum cliques, and maximum independent sets. We compare with three neural network based methods. We compare to two common approaches for backpropagating through discrete functions: the REINFORCE algorithm (Williams, 1992), and the Straight-Through estimator (Bengio et al., 2013). The third is the recently proposed probabilistic penalty relaxation (Karalias & Loukas, 2020) for combinatorial optimization objectives. All methods use the same GNN backbone, comprising a single GAT layer (Veličković et al., 2018) followed by multiple gated graph convolution layers Li et al. (2015).

In all cases, given an input graph $G = (V, E)$ with $|V| = n$ nodes, a GNN produces an embedding for each node: $\mathbf{X} \in \mathbb{R}^{n \times d}$. For scalar SFEs $d = 1$, while for neural SFEs we consider $\mathbf{X}\mathbf{X}^\top$ in order to produce an $n \times n$ PSD matrix, which is passed as input to the SFE $\mathfrak{F}$. The set function $f$ used is problem dependent, which we discuss below. Finally, see Appendix F for training and hyper-parameter optimization details, and Appendix E for details on data, hardware, and software.

**Maximum Clique.** A set $S \subseteq V$ is a clique of $G = (V, E)$ if $(i, j) \in E$ for all $i, j \in S$. The MaxClique problem is to find the largest set $S$ that is a clique: i.e., $f(S) = |S| \cdot \mathbf{1}\{S \text{ a clique}\}$.

**Maximum Independent Set (MIS).** A set $S \subseteq V$ is an independent set of $G = (V, E)$ if $(i, j) \notin E$ for all $i, j \in S$. The goal is to find the largest $S$ in the graph that is independent, i.e., $f(S) = |S| \cdot \mathbf{1}\{S \text{ an ind. set}\}$. MIS differs significantly from MaxClique due to its high heterophily.

**Results.** Table 1 displays the mean and standard deviation of the approximation ratio $f(S)/f(S^*)$ of the solver solution $S$ and an optimal $S^*$ on the test set graphs. The neural Lovász extension outperforms its scalar counterpart in 8 out of 10 cases, often by significant margins, for instance improving a score of $0.778$ on PROTEINS MaxClique to $0.926$. The neural SFE proved effective at boosting poor scalar SFE performance, e.g., $0.311$ on ENZYMES MIS, to the competitive performance of $0.775$. Neural Lovász outperformed or equalled and straight-through in 9 out of 10 cases, and the method of Karalias & Loukas (2020) in 6 out of 10.

## 5.2 Constraint Satisfaction Problems

Constraint satisfaction problems ask if there exists a set satisfying a given set of conditions (Kumar, 1992; Cappart et al., 2021b). In this section, we apply SFEs to the $k$-clique problem: given a graph, determine if it contains a clique of size $k$ or more. We test on the ENZYMES and PROTEINS datasets. Since satisfiability is a binary classification problem we evaluate using F1 score.

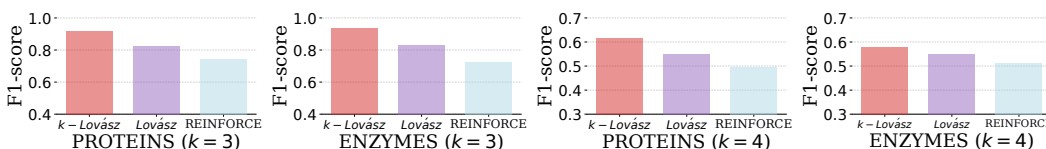

Figure 2: $k$-**clique constraint satisfaction:** higher F1-score is better. The $k$-bounded cardinality Lovasz extension is better aligned with the task and significantly improves over the Lovász extension.

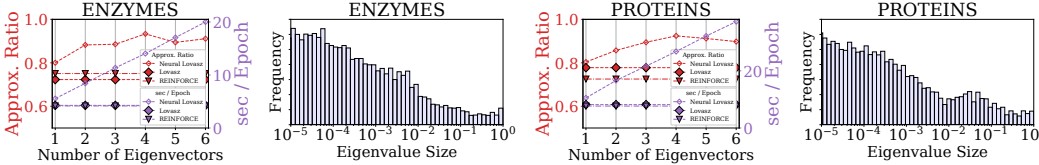

Figure 3: **Left:** Runtime and performance of neural SFEs on MaxClique using different numbers of eigenvectors. **Right:** Histogram of spectrum of matrix $\mathbf{X}$, outputted by a GNN trained on MaxClique.

**Results.** Figure 2 shows that by specifically searching over sets of size $k$ using the cardinality constrained Lovász extension from Section 3.1, we significantly improve performance compared to the Lovász extension, and REINFORCE. This illustrates the value of SFEs in allowing task-dependent considerations (in this case a cardinality constraint) to be built into extension design.

## 5.3 Training Error as a Classification Objective

During training the performance of a classifier $h$ is typically assessed using the training error $\frac{1}{n}\sum_{i=1}^{n}\mathbf{1}\{y_i \neq h(x_i)\}$. Since training error itself is non-differentiable, it is standard to train $h$ to optimize a differentiable surrogate such as the cross-entropy loss. Here we offer an alternative training method by continuously extending the non-differentiable mapping $\hat{y} \mapsto \mathbf{1}\{y_i \neq \hat{y}\}$. This map is a set function defined on single item sets, so we use the singleton extension (definition in Section 3.1). Our goal is to demonstrate that the resulting differentiable loss function closely tracks the training error, and can be used to minimize it. We do not focus on test time generalization. Figure 6 shows the results. The singleton extension loss (left plot) closely tracks the true train-

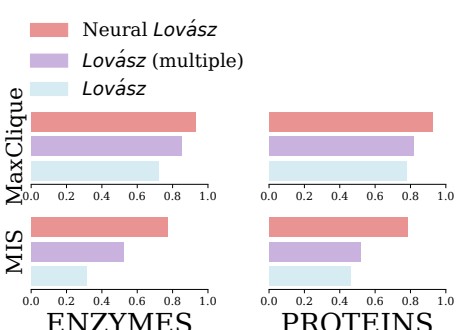

Figure 4: Neural SFEs outperform a naive alternative high-dimensional extension.

ing error at the same numerical scale, unlike other common loss functions (see Appendix G for setup details). While we leave further consideration to future work, training error extensions may be useful for model calibration (Kennedy & O'Hagan, 2001) and uncertainty estimation (Abdar et al., 2021).

## 5.4 Ablations

**Number of Eigenvectors.** Figure 3 compares the runtime and performance of neural SFEs using only the top-$k$ eigenvectors from the eigendecomposition $\mathbf{X} = \sum_{i=1}^{n} \lambda_i \mathbf{x}_i \mathbf{x}_i^\top$ with $k \in \{1,2,3,4,5,6\}$ on the maximum clique problem. For both ENZYMES and PROTEINS, performance increases with $k$—easily outperforming scalar SFEs and REINFORCE—until saturation around $k = 4$, while runtime grows linearly with $k$. Histograms of the eigenvalues produced by trained networks show a rapid decay in the spectrum, suggesting that the smaller eigenvalues have little effect on $\mathfrak{F}$.

**Comparison to Naive High-Dimensional Extension.** We compare neural SFEs to a naive high-dimensional alternative which, given an $n \times d$ matrix $\mathbf{X}$ simply computes a scalar SFE on each column independently and sums them up. This naive function design is not an extension, and the dependence on the $d$ dimensions is linearly separable, in contrast to the complex non-linear interactions between columns of $\mathbf{X}$ in neural SFEs. Figure 4 shows that this naive extension, whilst improving over one-dimensional extensions, performs considerably worse than neural SFEs.

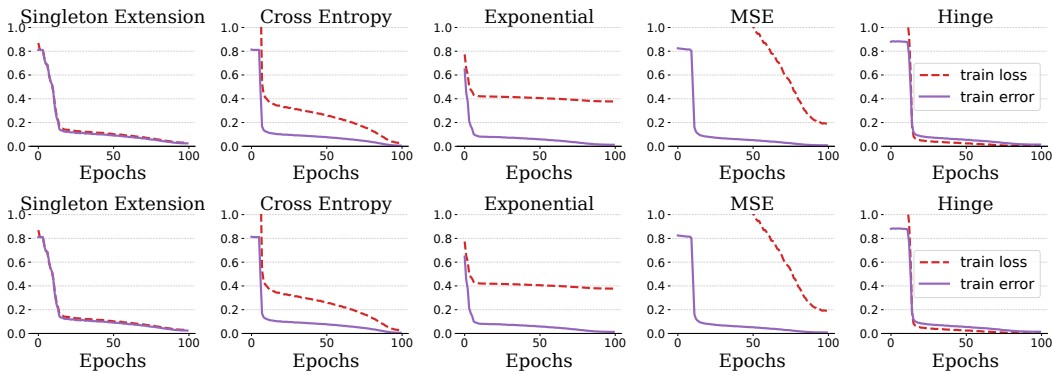

Figure 5: Top: CIFAR10. Bottom: SVHN. The singleton extension loss (left) is the only loss that approximates the true non-differentiable training error at the same numerical scale.

## 6 Related Work

**Neural combinatorial optimization** Our experimental setup largely follows recent work on unsupervised neural combinatorial optimization (Karalias & Loukas, 2020; Schuetz et al., 2022; Xu et al., 2020; Toenshoff et al., 2021; Amizadeh et al., 2018), where continuous relaxations of discrete objectives are utilized. In that context, it is important to take into account the key conceptual and methodological differences of our approach. For instance, in the unsupervised *Erdős goes neural* (EGN) framework from Karalias & Loukas (2020), the probabilistic relaxation and the proposed choice of distribution can be viewed as instantiating a multilinear extension. As explained earlier, this extension is costly in the general case (since $f$ must be evaluated $2^n$ times, and summed) but can be computed efficiently in closed form in certain cases. On the other hand, our extension framework offers multiple options for efficiently computable extensions without imposing any further conditions on the set function. For example, one could efficiently (linear time in $n$) compute the scalar and neural Lovász extensions of any set function with only black-box access to the function. This renders our framework more broadly applicable. Furthermore, EGN incorporates the problem constraints additively in the loss function. In contrast to that, our extension framework does not require any commitment to a specific formulation in order to obtain a differentiable loss. This provides more flexibility in modelling the problem, as we can combine the cost function and the constraints in various other ways (e.g., multiplicatively). For general background on neural combinatorial optimization, we refer the reader to the surveys (Bengio et al., 2021; Cappart et al., 2021a; Mazyavkina et al., 2021).

**Lifting to high-dimensional spaces.** Neural SFEs are heavily inspired by the Goemans-Williamson (Goemans & Williamson, 1995) algorithm and other SDP techniques (Iguchi et al., 2015), which lift problems onto higher dimensional spaces, solve them, and then project back down. Our approach to lifting set functions to high dimensions is motivated by the algorithmic alignment principle (Xu et al., 2019): neural networks whose computations emulate classical algorithms often generalize better with improved sample complexity (Yan et al., 2020; Li et al., 2020; Xu et al., 2019). Emulating algorithmic and logical operations is the focus of Neural Algorithmic Reasoning (Veličković et al., 2019; Dudzik & Veličković, 2022; Deac et al., 2021) and work on knowledge graphs (Hamilton et al., 2018; Ren et al., 2019; Arakelyan et al., 2020), which also emphasize operating in higher dimensions.

**Extensions.** Scalar SFEs use an LP formulation of the convex closure (El Halabi, 2018, Def. 20), a classical approach for defining convex extensions of discrete functions (Murota, 1998, Eq. 3.57). See Bach (2019) for a study of extensions of submodular functions. The constraints of our dual LP arise in contexts from global optimization (Tawarmalani & Sahinidis, 2002) to barycentric approximation and interpolation schemes in computer graphics (Guessab, 2013; Hormann, 2014). Convex extensions have also been used for combinatorial penalties with structured sparsity (Obozinski & Bach, 2012, 2016), and general minimization algorithms for set functions (El Halabi & Jegelka, 2020).

**Stochastic gradient estimation.** SFEs produce gradients for $f$ requiring only black-box access. There is a wide literature on sampling-based approaches to gradient estimation, for instance the REINFORCE algorithm (Williams, 1992) (i.e., score function estimator). However, sampling introduces noise which can cause unstable training and convergence issues, prompting significant

study of variance reducing control variates (Gu et al., 2017; Liu et al., 2018; Grathwohl et al., 2018; Wu et al., 2018; Cheng et al., 2020). SFEs can avoid sampling (and noise) all-together, as our extensions are differentiable and can be computed deterministically. A closely related, yet distinct, task is to produce gradients through sampling operations, which introduce non-differentiable nodes in neural network computation graphs. The Straight-Through Estimator (Bengio et al., 2013), arguably the simplest solution, treats sampling as the identity map in the backward pass, yielding biased gradient estimates. The Gumbel-Softmax trick (Maddison et al., 2017; Jang et al., 2017), provides an alternative method to sample from categorical distributions (also benefiting from variance reduction (Paulus et al., 2020a)). The trick can be seen through the lens of the more general Perturb-and-MAP framework that treats sampling as a perturbed optimization program. This framework has since been used to generalize the trick to more complex distributions (Paulus et al., 2020b) and to differentiate through the parameters of exponential families for learning and combinatorial tasks (Niepert et al., 2021). Broadly, these techniques relax a discrete distribution into a continuous one by utilizing a noise distribution and *assuming access* to a continuous loss function. SFEs are complementary to this setup, addressing the problem of designing continuous extensions.

**Differentiating through convex programs and algorithms.** Recent years have seen a surge of interest in combining neural networks with solvers (e.g., LP solvers) and/or algorithms in differentiable end to end pipelines (Agrawal et al., 2019; Amos & Kolter, 2017; Paulus et al., 2021; Pogančić et al., 2019; Wang et al., 2019). Whilst sharing the algorithmic alignment motivation of SFEs, the convex programming connection is mostly cosmetic: these works directly embed solvers into network architectures, while SFEs use convex programs as an analytical tool, without requiring solver access.

## 7 Conclusion

We introduced Neural Set Function Extensions, a framework that enables evaluating set functions on continuous and high dimensional representations. We showed how to construct such extensions and demonstrated their viability in a range of tasks including combinatorial optimization and image classification. Notably, neural extensions deliver good results and improve over their scalar counterparts, further affirming the benefits of problem-solving in high dimensions.

## 8 Acknowledgements

NK would like to thank Marwa El Halabi, Mario Sanchez, Mehmet Fatih Sahin, and Volkan Cevher for the feedback and fruitful discussions. NK and AL would like to thank the Swiss National Science Foundation for supporting this work in the context of the project "Deep Learning for Graph-Structured Data" (grant number PZ00P2179981). SJ and JR acknowledge support from NSF CAREER award 1553284, NSF award 1717610, and the NSF AI Institute TILOS.

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
