# OpenReview forum: "Neural Set Function Extensions: Learning with Discrete Functions in High Dimensions"
_NeurIPS.cc/2022/Conference — NeurIPS 2022 Accept_

### Official Review · Reviewer_Wk2P · 2022-07-08

**Rating:** 8
**Confidence:** 2
**Soundness:** 3 good
**Presentation:** 4 excellent
**Contribution:** 4 excellent

**Summary:**

The paper is about training neural networks on discrete objects, and sets in particular. The set loss functions are typically non-differentiable, but the authors propose a method for extending the space of discrete set embeddings to continuous space. This means we can evaluate set loss functions on continuous outputs of a neural network, enabling training on non-differentiable losses. The extended set function is defined as a convex combination of the set function values at discrete points. The authors propose several ways of setting the weights of the convex combination and choosing which sets the function is evaluated on. The authors then extend the method to _matrix_ set embeddings (as opposed to vector embeddings), arguing that the increased dimensionality aids learning. The authors evaluate the method on a set of discrete optimization tasks.

**Questions:**

- Have authors measured the runtimes/memory requirements of the various variants of the proposed method? For example, does the runtime/memory use increase significantly when using neural SFE as opposed to the scalar SFE? How do both compare to REINFORCE/Erdos?

- Consider re-naming figures 3-5 to have them go in order.

**Limitations:**

I believe the discussion on the limitations of the method could be extended (see the weaknesses and questions above).

Authors do not discuss potential negative societal impact of their work, but I consider them to be minor, albeit methods like these could potentially be used for mining social graphs.

**Strengths And Weaknesses:**

Strengths:
- A contribution to the important field of neural network learning in discrete spaces.
- The paper is written well: the narrative is clear, the mathematical notation is crisp, the discussion of related literature is extensive.
- Original work with a great deal of technical complexity: the authors lean on the discrete optimization literature, but adapt the ideas in the neural network learning context.
- The theoretical motivation of the method is sound, and the empirical evaluation is extensive.
- The method consistently beats a strong baseline (REINFORCE).

Weaknesses:
- Even though neural SFE is consistently beating the scalar SFE, the results of both methods are less impressive when compared to the strongest baseline (Erdos).
- Given the previous point, a deeper discussion of the benefits/drawbacks of Erdos when compared to SFE and a more extensive empirical comparison of the two methods would make for a stronger contribution.

---

> ### Author Response · Authors · 2022-08-02
> **Response to points raised in the review**
>
> We are grateful for the positive feedback and the comments on our work, and we are happy about your appreciation of the originality and the technical contribution of the paper. We are  glad to answer your clarifying questions, which we will also add to the manuscript as they will improve the clarity of our work for other readers too.
>
>  ---
>
> >  [Could you discuss] the benefits/drawbacks of Erdős when compared to SFE and [give] a more extensive empirical comparison?
>
> **SFEs are a more general framework**
> - (Neural) SFEs rely on a combination of weighted evaluations of the set function which can be computed efficiently under the choice of a distribution with small support (e.g., the Lovász Extension with a distribution over a chain of $n$ sets). The Erdős method assumes that it is possible to compute the expectation $F = \mathbb{E}_S [f(S)]$ where $S$ is sampled from a product of Bernoulli distributions—which has support on all $2^n$ possible subsets of the ground set—which takes $O(2^n)$ time in general. This makes SFEs more practical in the general case.
> - Since an exponential-size sum is not tractable, the Erdős method relies on special properties of $f$ to compute a closed form expression for $F$. This is only possible in some scenarios (e.g., max clique and max independent set, or in general linear programs), but not for arbitrary constraints. For instance it is not straightforward to derive such an expectation for a function that determines whether a subgraph is connected (this is related to the network reliability problem, known to be NP-hard).
> - SFEs are more general than Erdős in the literal sense that Erdős is a special case where a particular choice of distribution over sets is chosen (a product of Bernoulli distributions). Furthermore, Erdős imposes the use of a specific formulation for constrained optimization where the constraints are additively included in the loss with a penalty term. SFEs do not have such requirements.
> - Since SFEs allow many other choices of distributions over sets, they can be adapted to more problems. For instance, by choosing a distribution supported only on sets of at most size $k$ we show that SFEs can be tailored to the problem of detecting $k$-cliques (Fig 2).  Additionally, certain discrete functions are better matched with particular distributions. For example, it is known that the Lovász Extension will always be convex if and only if the discrete function is submodular. Thus, having the ability to choose the extension based on prior knowledge about the function can be helpful in terms of optimization.
> - SFEs can be applied to problems outside of the field of combinatorial optimization, like our experiments on directly optimising training error for image classifiers. We think this illustrates the promise and wider applicability of SFEs.
>
>
> **SFEs require only black-box access to the set function $f$**
> - In contrast to Erdős, which, when implemented efficiently, must be able to derive expectations over $f$, SFEs only require black box access to $f$.
> - By choosing the right distribution over sets—e.g,. as in the Lovász extension—SFEs can always be computed efficiently in $O(n)$ time.

---

> > ### Comment · Reviewer_Wk2P · 2022-08-09
> > **Re: Response to points raised in the review**
> >
> > I thank the authors for clarifying these points: including the extended discussion of Erdos will make the paper more convincing, as reinforced by the discussion with the other reviewer. The authors did not address my question on runtime/memory requirements, but I do encourage them to discuss these aspects in the final version, which would make the paper more appealing to practitioners. In the end, I maintain my strong recommendation for accepting the paper.

---

> > > ### Author Response · Authors · 2022-08-09
> > > **Response**
> > >
> > > We apologise for neglecting to provide more details regarding runtime and memory costs. Here are some clarifications:
> > >
> > > - For runtime, scalar SFEs are comparable to the other methods (Erdos and REINFORCE).
> > > Neural SFEs have essentially a $k$-fold slowdown compared to scalar SFEs, where $k$ is some small number typically around $3$. The quantity $k$ represents the number of eigenvectors used in the neural SFE. In Figure 3, precise times and how they relate to performance can be found. We found that the extra time cost for small $k$ has a favorable performance/time tradeoff but the benefits diminish for larger values of $k$.
> > >
> > > - For memory costs, there is no special overhead imposed by our method compared to other methods. This is also the case when comparing neural SFEs to scalar ones.  All experiments could fit comfortably in a single GPU without any memory issues.
> > >
> > > We thank the reviewer for taking the time to respond to our comments. We'll make sure to highlight the points that were brought up in the relevant sections of the paper.

---

### Official Review · Reviewer_NU5a · 2022-07-11

**Rating:** 6
**Confidence:** 2
**Soundness:** 3 good
**Presentation:** 2 fair
**Contribution:** 3 good

**Summary:**

This paper proposes a framework for extending functions on discrete sets to continuous domains. They propose a method called "neural set function extensions" for approximating these function extensions with neural networks in a way that is amenable to standard gradient-based optimization techniques. The proposed framework is evaluated on a number of discrete combinatorial optimization problems such as finding maximum cliques in a graph on a number of existing graphical benchmarks.

**Questions:**

Can you give a clearer overview of the relevance of the problem area? Where is neural optimization of discrete graphical properties such as maximum cliques likely to be relevant to the broader community? Do existing models in the literature fail to address these cases? The results in table 1 don't seem to show that you convincingly outperform existing baselines - is there another way in which your proposed model is superior to existing methods?

The k-clique constraint satisfaction experiments did not compare against the Erdos model from table 1 - is this model not applicable for these experiments, or was there another reason it was omitted?



**Limitations:**

This is a technical paper and is unlikely to have any obvious negative social impacts.

As mentioned in previous sections, I think the paper would benefit from a clearer discussion of where the proposed method is applicable and why these applications are significant.

**Strengths And Weaknesses:**

This paper proposes an interesting and novel framework for embedding functions on discrete sets into a differentiable framework that can be optimized using standard gradient-based techniques. The proposed techniques are mathematically interesting, and may allow neural models to be applied to regimes that were previously very difficult to optimize.

My primary concern with this work is with the significance of the proposed methods. It is unclear to me how relevant the problem settings addressed by the authors are, and whether they are addressed satisfactorily by existing methods. I think the paper would benefit from a clearer discussion of where these models could be applied, and how these applications are relevant to the wider machine learning community, as well as the degree to which existing methods fail in these regimes.
The empirical results in the paper also seem rather weak - on the maximum clique and maximum independent set experiments shown in table 1, the proposed model only outperformed existing baselines by statistically significant margins on 1 category out of 10 - in all others it was worse or within standard deviation of the baseline. On the maximum independent set task, the proposed model performed worse than the baselines on every dataset, sometimes by large margins.
All in all, I find the mathematical framework proposed by the paper to be quite interesting, but I am not entirely convinced by the significance of the contribution.

---

> ### Author Response · Authors · 2022-08-02
> **Response to points raised in the review: Part 1/2**
>
> We would like to thank the reviewer for taking the time to review our paper and provide valuable feedback. We are glad that you found SFEs to be “an interesting and novel framework…. [that] may allow neural models to be applied to regimes that were previously very difficult to optimize.” These comments recognize important contributions we make, and we hope can be a solid foundation to eventually viewing our work as worthy of strong acceptance.
>
> The reservations mentioned in your review focus on the relevance of the problem area, and experimental comparisons. We clarify these by giving a clearer discussion of the wider context and motivation behind our work. We will update the manuscript to reflect this discussion since it will also be important to other readers of our paper.
>
> ---
>
> > Can you give a clearer overview of the relevance of the problem area? Where is neural optimization of [combinatorial problems] likely to be relevant to the broader community?
>
> **Neural combinatorial optimization is a rapidly developing area.**
> - Neural combinatorial optimization is a nascent field that has attracted interest in the deep learning community [1,5] with high profile publications that have garnered mainstream attention (e.g., see the recent work by Google on chip design [4]).
> Other notable works include the recent paper that just appeared in Nature [4] for QUBO problems. For more information on applications and the breadth of the topic we recommend well known surveys from the field [1,2].
>
>
> **Integrating combinatorial problem solving in neural pipelines can have significant impact in AI research.**
> - We chose to consider maximum independent set and maximum clique since they are _fundamental_ combinatorial problems with long histories and numerous applications in the fields of science and engineering [12,13]. We therefore think these problems are a suitable testbed for demonstrating the promise of SFEs.
> - Implementing systems that can solve more complex combinatorial problems (e.g., games like chess) requires significant engineering effort that is beyond the scope of the paper. However, we believe that SFEs can be a useful addition to the toolset of engineers working on those problems by facilitating the use of discrete functions with continuous end-to-end optimization methods like gradient descent.
> - Puzzles and games like Sudoku can be directly mapped to combinatorial problems like SAT. More complex games like Go or Poker have strong ties to combinatorics and have been extensively studied in the machine learning community [6,7].
> - Tasks like sorting and ranking which are of utmost importance for data science and machine learning are also combinatorial in character [9].
> - Structured prediction is also another subfield of machine learning with applications to natural language and reasoning tasks, which has strong ties to combinatorial optimization [8].
>
> **Neural networks can contribute to improvements in the state of the art for combinatorial optimization.**
> - Conversely, neural networks can be used to improve existing methods [11] and may offer promising tools for advancing the state of the art in combinatorial optimization (e.g., [4,10]). Our extensions describe a new framework towards that goal that can help tackle a larger variety of problems.
> ---
>
> **References (publication venues omitted due to character limit)**
>
> [1] Bengio, Yoshua, Andrea Lodi, and Antoine Prouvost. "Machine learning for combinatorial optimization: a methodological tour d’horizon."
>
> [2] Cappart, Quentin, et al. "Combinatorial optimization and reasoning with graph neural networks."
>
> [3] Mirhoseini, Azalia, et al. "A graph placement methodology for fast chip design."
>
> [4] Schuetz, Martin JA, J. Kyle Brubaker, and Helmut G. Katzgraber. "Combinatorial optimization with physics-inspired graph neural networks."
>
> [5] Vinyals, Oriol, Meire Fortunato, and Navdeep Jaitly. "Pointer networks."
> [6] Brown, Noam, and Tuomas Sandholm. "Superhuman AI for heads-up no-limit poker: Libratus beats top professionals."
>
> [7] Silver, David, et al. "Mastering the game of go without human knowledge."
>
> [8] Niculae, Vlad, et al. "Sparsemap: Differentiable sparse structured inference."
>
> [9] Blondel, Mathieu, et al. "Fast differentiable sorting and ranking."
>
> [10] Ahn, Sungsoo, Younggyo Seo, and Jinwoo Shin. "Learning what to defer for maximum independent sets."
>
> [11] Selsam, Daniel, and Nikolaj Bjørner. "Guiding high-performance SAT solvers with unsat-core predictions."
>
> [12] Malod-Dognin, Noël, Rumen Andonov, and Nicola Yanev. "Maximum cliques in protein structure comparison."
>
> [13] Shirokikh, Oleg, Vladimir Stozhkov, and Vladimir Boginski. "Combinatorial optimization techniques for network-based data mining."
>
> ---
>  **Continued in the next comment**

---

> > ### Author Response · Authors · 2022-08-02
> > **Response to points raised in the review: Part 2/2**
> >
> > > On the $k$-clique constraint satisfaction experiments why didn’t you compare against the Erdős model from Table 1?
> >
> >
> >
> > **The goal of the experiment is to show that SFEs can incorporate problem knowledge.**
> > - The takeaway of this experiment is that SFEs are flexible, and can build in problem knowledge (in this case a cardinality constraint) to improve performance. This illustrates the versatility of SFEs, and was not intended as a performance benchmark between SFEs and other methods. We include REINFORCE as a simple baseline just to sanity check that the problem is non-trivial.
> > - Furthermore, the Erdős method is presented as an approach to constrained optimization problems while k-clique is a constraint satisfaction problem. Since the Erdős paper does not offer a methodology for constraint satisfaction, we opted for REINFORCE which can be directly applied in this setting.
> >
> > ---
> > > The results in table 1 don't seem to show that you convincingly outperform existing baselines - is there another way in which your proposed model is superior to existing methods?
> >
> >
> > **Yes: our method is more widely applicable than the Erdős method.**
> > - SFEs overcome certain limitations of the Erdős method which allows them to be immediately applied to the many problems that Erdős cannot be adapted to so simply. Examples given in our paper include detecting $k$-cliques, and using SFEs to train image classifiers. SFEs can be computed efficiently in $O(n)$ time for *any* set function with only black-box access. In contrast, the Erdős method introduces a loss that is computable in $O(2^n)$ time in general (where $n$ is the size of the ground set) and can be computed efficiently only in special cases (e.g., linear programs).
> >
> >
> >
> > **REINFORCE is also widely applicable, but performs worse than SFEs.**
> > - SFEs and REINFORCE are both generally applicable to many problems with minimal assumptions (both assume only black-box access to set function $f$). While both have this advantage over the Erdős method, SFEs have an empirical performance advantage over REINFORCE: our experiments generally find SFEs to perform better.
> > - It should be noted that the standard deviation is reported over the graphs in the test set and not over repeated runs over the same data. The reported number is chosen after a grid search over multiple parameter configurations. We found that the performance of the best model after re-runs of the hyperparameter sweep was very consistent.

---

> > > ### Comment · Reviewer_NU5a · 2022-08-08
> > > **Response to authors**
> > >
> > > Thank you for the clarification. This does address some of my concerns with the paper. If the Erdos model is indeed highly limited in applicability then the empirical results look far more convincing.

---

> > > > ### Author Response · Authors · 2022-08-09
> > > > **Response**
> > > >
> > > > "Thank you very much for your consideration of our response, and for adjusting your score!"

---

### Author Response · Authors · 2022-08-02
**Response to all reviewers**

We would like to thank the reviewers for taking the time to read our paper and provide valuable comments on our work. The feedback will help us improve the clarity of the paper and we will subsequently update the relevant sections once given the additional space for the camera-ready version. We believe our rebuttal addresses the reservations mentioned in the reviews.

We are happy to answer any further questions regarding the paper and provide explanations if needed.

---

### Meta-Review · Area_Chair_pDik · 2022-08-25

**Recommendation:** Accept
**Confidence:** Certain

**Metareview:**

This paper proposes a new neural set function extensions methodology that allows approximating function on sets with neural networks in a way that is amenable to standard gradient-based optimization techniques. The proposed methodology is evaluated on a number of discrete combinatorial optimization problems such as finding maximum cliques in a graph on a number of existing graphical benchmarks.

Allowing neural networks (and other ML methods) to natively handle discrete functions is important in a variety of applications. This paper is a step towards a more robust solution to this problem and a worthy contributions to the conference.

**Award:**

No

---

### Decision · Program_Chairs · 2022-09-14

Accept